# Influence of Temperature Parameters on Morphological Characteristics of Plasma Deposited Zinc Oxide Nanoparticles

**DOI:** 10.3390/nano12111838

**Published:** 2022-05-27

**Authors:** Tatyana Sergeevna Sazanova, Leonid Alexandrovich Mochalov, Alexander Alexandrovich Logunov, Mikhail Alexandrovich Kudryashov, Diana Georgievna Fukina, Maksim Anatolevich Vshivtsev, Igor Olegovich Prokhorov, Pavel Andreevich Yunin, Kirill Alexandrovich Smorodin, Artem Anatolevich Atlaskin, Andrey Vladimirovich Vorotyntsev

**Affiliations:** 1Laboratory of Membrane and Catalytic Processes, Nanotechnology and Biotechnology Department, Nizhny Novgorod State Technical University n.a. R.E. Alekseev, Minin Str. 24, 603950 Nizhny Novgorod, Russia; an.vorotyntsev@gmail.com; 2Chemical Engineering Laboratory, Research Institute for Chemistry, Lobachevsky State University of Nizhny Novgorod, Gagarin Ave. 23, 603022 Nizhny Novgorod, Russia; mochalovleo@gmail.com (L.A.M.); alchemlog@gmail.com (A.A.L.); mikhail.kudryashov1986@yandex.ru (M.A.K.); dianafuk@yandex.ru (D.G.F.); mvshivtcev@mail.ru (M.A.V.); igorprokhorov1998@yandex.ru (I.O.P.); smorodin.kirill.a@gmail.com (K.A.S.); 3Department for Technology of Nanostructures and Devices, Institute for Physics of Microstructures of the Russian Academy of Science, Academic Str. 7, Afonino, 603087 Nizhny Novgorod, Russia; yunin@ipmras.ru; 4Laboratory of SMART Polymeric Materials and Technologies, Mendeleev University of Chemical Technology, Miusskaya Sq. 9, 125047 Moscow, Russia; atlaskin.a.a@muctr.ru

**Keywords:** zinc oxide, nanoparticles, PECVD, structure, morphology

## Abstract

Zinc oxide nanoparticles were obtained by plasma-enhanced chemical vapor deposition (PECVD) under optical emission spectrometry control from elemental high-purity zinc in a zinc–oxygen–hydrogen plasma-forming gas mixture with varying deposition parameters: a zinc source temperature, and a reactor temperature in a deposition zone. The size and morphological parameters of the zinc oxide nanopowders, structural properties, and homogeneity were studied. The study was carried out with use of methods such as scanning electron microscopy, X-ray structural analysis, and Raman spectroscopy, as well as statistical methods for processing and analyzing experimental data. It was established that to obtain zinc oxide nanoparticles with a given size and morphological characteristics using PECVD, it is necessary (1) to increase the zinc source temperature to synthesize more elongated structures in one direction (and vice versa), and (2) to decrease the reactor temperature in the deposition zone to reduce the transverse size of the deposited structures (and vice versa), taking into account that at relatively low temperatures instead of powder structures, films can form.

## 1. Introduction

Currently, nanosized zinc oxide is one of the important materials in development for various fields of medicine and industry. Zinc oxide is a unique material because of its advantages such as cost efficiency and variable engineering properties, good biocompatibility and antibacterial properties, adjustable band-gap and particle size/shape, and many other features [1], making it applicable in a wide range of fields. In particular, zinc oxide nanoparticles can be used as: excellent antibacterial, antioxidant, antidiabetic and tissue regenerating agents [2]; innovative anticancer agents [3]; material for photocatalytic degradation of organic pollutant [4,5]; material for manufacture of electronic devices over flexible substrates [6]; an electron transport layer for quantum dot light-emitting diodes [7]; nanocomposite electrode material for supercapacitor [8]; and material for the production of gas sensors [9]. 

Nanosized zinc oxide is obtained in the form of rods (threads), ridges, honeycombs, rings, ribbons, springs (spirals), cells, tetrapods, as well as thin films and coatings. For the synthesis of all these numerous modifications, numerous preparation methods were proposed, of which two main groups can be distinguished: physical (sputtering, laser ablation, electrospraying, ball milling, electron beam evaporation, etc.) and chemical (microemulsion technique, sol–gel method, co-precipitation method, hydrothermal method, polyol method, chemical vapor deposition, etc.). The main advantage of the chemical methods is the possibility of obtaining particles with a given size, composition, and structure [10]. This is important because the shape and size of nanoparticles directly affects their properties. For example, Hammad T. and co-workers [11] reported changing a red shift from 3.62 to 3.33 eV when the average particle size of ZnO nanoparticles was increased from 11 to 87 nm, respectively. In another work, Mornani E. and co-workers [12] reported a change in the band gap from 4.45 to 4.08 eV with an increase in the average size of ZnO nanoparticles from 46 to 66 nm, respectively. Similarly, controlling nanoparticle sizes to determine their optimal range is significant for mechanical properties of nanoparticle assemblies [13], the rate of gas photofixation using powder catalysts based on nanoparticles [14], and others [15,16,17,18,19].

One of the most promising methods for producing zinc oxide nanoparticles of a given size and shape is plasma-enhanced chemical vapor deposition (PECVD). Plasma initiation makes it possible to significantly reduce the temperature of reactor walls and a deposition zone, as well as to eliminate the pollution possibility of the final product with equipment materials and to control the deposition zone temperature over a wider range, thereby setting the conditions for structure growth. Plasma initiation also makes it possible to achieve 100% conversion of initial substances due to establishing kinetic dependencies during plasma–chemical reactions. Thus, PECVD provides controllability, one stage, cost-effectiveness, high purity of resulting materials, as well as versatility and process scalability [20,21,22]. However, to widely use PECVD in the preparation of nanosized zinc oxide with given morphology, it is necessary to fundamentally study the influence of process parameters on the resulting product.

This work is devoted to studying zinc oxide nanoparticles obtained by the PECVD method from elemental high-purity zinc in a zinc–oxygen–hydrogen plasma-forming gas mixture with varying deposition parameters, namely a zinc source temperature and a reactor temperature in a deposition zone. 

## 2. Experimental Section

### 2.1. Materials

Zinc of 5N purity (Changsha Rich Nonferrous Metals Co Ltd., Changsha, Hunan, China), high purity hydrogen (99.9999%) and oxygen 6.0 (99.9999%) (Horst Technologies Ltd., Dzerzhinsk, Nizhny Novgorod region, Russia) were used as components of a plasma-forming mixture. Hydrogen was used as a carrier gas, which also acted as a temperature stabilizer and regulator of the nanostructure growth.

### 2.2. Plasma–Chemical Synthesis

The scheme of a plasma–chemical installation is shown in Figure 1. The installation consisted of a gas supply system, a pumping system, a cuvette with initial zinc, and a pear-shaped plasma–chemical reactor made of high-purity quartz glass with a total volume of about 1200 cm^3^. An external inductor, an RF generator (with an operating frequency of 40.68 MHz and a maximum power of 500 W), and a universal matching device were used to ignite an inductively coupled or mixed nonequilibrium plasma discharge.

A loading cuvette in the form of a boat with metal zinc was placed in a furnace with an external resistive heater and an internal thermocouple. A tank for collecting zinc oxide powder was placed into the reactor through a stainless-steel vacuum loading flange and was installed perpendicular to the carrier gas flow on a special movable holder. The loading cuvette, the powder tank, and the movable holder were made of high-purity quartz glass. To cool the reactor in the deposition zone, a circulation thermostat was used.

High-pure granulated Zn was loaded into the quartz furnace. High-pure H_2_ was blown through the Zn source with the constant rate of 15 mL/min. The vapors of Zn were delivered by the career gas via the quartz lines heated up to 550 °C with the internal diameter of 6 mm directly into the plasma zone, where the formation of ZnO materials took place. The temperature of the lines and external surface of the plasma chamber was measured by a pyrometer. The temperature of the substrate holder was found by the internal thermocouple. High-pure O_2_ was fed from below towards the main carrier gas with zinc vapor directly in the plasma zone with the constant rate of 15 mL/min. The total gas flow through the plasma–chemical reactor was set equal to 30 mL/min at a total pressure in the system of 0.1 Pa. The duration of each individual experiment (with the various deposition parameters) was one hour.

Before experiments, the installation was evacuated to a pressure of 1 × 10^−3^ Pa for several hours to remove traces of nitrogen and water from the walls of the reactor. Then, the powder tank was closed with a magnetic diaphragm of a special design.

The installation was also equipped with a high-resolution optical emission spectrometer HR4000CJ-UV-NIR (Avantes, The Netherlands) operating in the range of 180–1100 nm to control the excited intermediate species in the gas phase during the plasma–chemical process. The upper part of the plasma chamber was equipped with two plane-parallel windows made of special quartz glass of high transparency, maintained just after the inductor where the intensity of the lines was maximal. 

According to the optical emission spectroscopy (OES) data (Figure 2), the composition of the plasma-forming mixture in all experiments included Zn (I), O (I), O (II), H (I), OH (I). Zn (II) emission lines were not observed.

### 2.3. Scanning Electron Microscopy and Energy-Dispersive X-ray Spectroscopy

The size-morphological characteristics of the zinc oxide samples were studied by scanning electron microscopy (SEM) using an electron microscope JSM-IT300LV (JEOL, Peabody, MA, USA) with an electron probe diameter of about 5 nm and a probe current of less than 0.5 nA (the operating voltage was 20 kV). SEM scanning was performed using low-energy secondary electrons and backscattered electrons under a low vacuum to eliminate the charge. As sample preparation for SEM, zinc oxide powders were applied onto carbon double-sided conductive tapes.

The size of the structures observed on SEM images was measured as the maximal diameter of their cross-section. For additional control of the measurement results, analysis of the images (determination of the average size of equivalent disk (D_avg_) of structures’ cross-section) was carried out using the method of watershed segmentation by a software SPMLab™ v5 (TopoMetrix, Santa Clara, CA, USA).

### 2.4. X-ray Structural Analysis

The structure of the zinc oxide samples was studied using X-ray diffraction (XRD) analysis on an X-ray diffractometer D8 Discover (Bruker, Germany) equipped with a sealed CuKα radiation source tube and a position-sensitive detector LynxEye. Diffraction patterns were obtained by θ/2θ scanning in the 2θ range of 10–66° with a step of 0.1°. 

As sample preparation for the XRD analysis, a small amount of zinc oxide powders was placed in the center of a quartz disk. Then, about 3 drops of distilled water were added to the sample and it was spread to a thin layer with a glass rod. Next, the sample was placed in a desiccator to dry before the XRD analysis.

The results obtained were compared with the database PDF-2 Release 2011, namely PDF 01-071-6424 for ZnO and PDF 00-004-0831 for Zn.

### 2.5. Raman Spectroscopy

Raman spectra were studied on a spectroscopy complex NTEGRA Spectra Raman (NT-MDT, Moscow, Russia) using a laser with a wavelength of 473 nm. The radiation was focused by a 20× objective lens with an aperture of 0.45. The laser spot diameter was 5 μm. The power of the unfocused laser radiation was controlled by a silicon photodetector 11PD100-Si (Standa Ltd., Vilnius, Lithuania) and varied in the range from 1 mW to 1 μW. The Raman spectra analysis of the samples was carried out according to the scheme for reflection in the frequency range 80–800 cm^−1^ with a resolution of 0.7 cm^−1^. As sample preparation for the Raman spectra analysis, zinc oxide powders were applied onto carbon double-sided conductive tapes. The measurements were carried out at room temperature.

### 2.6. Varying Temperature Parameters in Plasma–Chemical Deposition

To study the influence of a zinc source temperature on the morphology of ZnO nanoparticles, the temperature was varied from 370 to 470 °C. At the same time, the reactor temperature in the deposition zone was maintained at 250 °C. The plasma discharge power was 50 W. 

To study the influence of a reactor temperature on the morphology of ZnO nanoparticles, the temperature was varied from 25 to 350 °C. At the same time, the zinc source temperature was maintained at 420 °C. The plasma discharge power was 50 W.

## 3. Results and Discussion

### 3.1. Zinc Source Temperature and Morphology of ZnO Nanoparticles

As a result of the plasma–chemical synthesis, ZnO nanoparticles with different shapes were obtained. According to the SEM images (Figure 3), the observed nanoparticles assumed a sphere-like, columnar, and rod-like shape depending on the zinc source temperature.

According to statistical processing of the SEM data (the sample size was 100 measurements), it was found that the average size (in the cross-section) of the deposited particles changed with an increase in the zinc source temperature, along with a variation in their shape (Figure 4).

The average size of the ZnO structures decreased from 120 to 100 nm with a simultaneous decrease in the coefficient of variation from 46 to 44% caused by an increase in temperature from 370 to 420 °C. Next, the average size of the structures approached 45 nm with a coefficient of variation of 32% caused by a further increase in temperature to 470 °C. Such a change in the transverse size of the nanoparticles can be associated with the changeable mechanism of their growth and the corresponding redistribution of the Zn and O atoms in the spherical and elongated crystal structures.

The XRD analysis results for the ZnO structures obtained by the PECVD method at the various zinc source temperatures are shown in Figure 5. The diffraction peaks corresponding to (002), (101), (102), (103), (112), (201), (004), (202) planes are characteristic of a ZnO structure [16,23,24,25,26,27,28,29]. Such a diffraction pattern corresponds to a hexagonal structure of the wurtzite type. [23,24,29]. No other peaks associated with impurities were observed, indicating that high-purity ZnO nanoparticles were obtained. However, the diffraction peaks corresponding to (101) and (102) planes, which match with that of pure metallic Zn [30], were observed after increasing the zinc source temperature from 420 to 470 °C (Figure 5c). This means that there was an excess of Zn in this sample.

It should be noted that the XRD characterization revealed a strong preferred (002) orientation for all the obtained ZnO structures, indicating that the c-axis of the unit cell was aligned perpendicular to the horizontal plane of the deposition zone [28]. However, the (201) orientation became pronounced for the sample obtained at the source temperature of 420 °C (Figure 5b) and was slightly reduced with the increase in temperature from 420 to 470 °C (Figure 5c). Such a reorientation was probably related to the transition of the ZnO structures from spherical to elongated forms. 

The Raman spectra for the ZnO structures obtained at the various zinc source temperatures are shown in Figure 6. The peak at about 435 cm^−1^ is typical for ZnO structures, while the peaks at about 275 cm^−1^ and 580 cm^−1^ do not correspond to ZnO normal modes. These peaks are additional vibrational modes and can be associated with defects and bond breaking, respectively [31,32,33]. 

Nevertheless, one must take into account the fact that the shape of Raman spectra depends on crystal orientation. In the case under consideration, based on the XRD data (Figure 5), the preferred orientation of the nanoparticles for all samples was (002). Moreover, for the sample obtained at the zinc source temperature of 420 °C (Figure 5b), the (201) orientation became pronounced, and the peak intensities in the Raman spectra (Figure 6) at about 275 cm^−1^ and 580 cm^−1^ had average values relative to other samples. The maximum intensities of these peaks were reached in another sample obtained at the source temperature of 470 °C, when the height of the (201) reflection on the XRD pattern (Figure 5c) was already reduced. Thus, the changes in the form of the Raman spectra can really be associated with defects and bond breaking in the case under consideration.

Analyzing the obtained Raman spectra, it can be concluded that the defectiveness of the ZnO structures (the peak at about 275 cm^−1^) increases with the rise in the zinc source temperature. The degree of bond breaking (the peak at about 580 cm^−1^) also has similar temperature dependence. Such a trend can be associated with increasing the Zn excess in the obtained ZnO structures, which can be led by bond breaking, and thereby causes the formation of defects similar to interstitial Zn.

### 3.2. Reactor Temperature and Size Distribution of ZnO Nanoparticles

As a result of the ZnO plasma–chemical synthesis at the reactor temperature in the deposition zone of 25 °C, a planar structure was formed instead of a nanoparticle one (Figure 7). At higher temperatures of the reactor deposition zone (250 and 350 °C), ZnO columnar structures were formed. Moreover, the average diameter (in cross-section) of the observed structures increased with a rise in the temperature.

According to statistical processing of the SEM data (the sample size was 100 measurements), it was established that the transverse diameter (at the widest part) of the ZnO columnar structures increased by three times with an increase in the reactor temperature in the deposition zone from 250 to 350 °C (Figure 8), and the coefficient of variation for the size range decreased from 44 to 27%.

The XRD analysis results for the ZnO structures obtained by the PECVD method at the various temperatures in the reactor deposition zone are shown in Figure 9. The diffraction peaks corresponding to (002), (101), (103), (112), (201), (004), and (202) planes are characteristic of a ZnO structure [16,23,24,25,26,27,28,29]. Such a diffraction pattern corresponds to a hexagonal structure of the wurtzite type [23,24,29]. No other peaks associated with impurities were observed, indicating that the high-purity ZnO nanoparticles were obtained. 

The XRD characterization revealed a strong preferred (002) orientation for all the obtained ZnO structures, indicating that the c-axis of the unit cell was aligned perpendicular to the horizontal plane of the deposition zone [28]. Notably, the diffraction pattern of the sample obtained at the temperature in the reactor deposition zone of 25 °C included only two pronounced peaks, namely (002) and (004). Such a pattern is characteristic of ZnO films [26]. Moreover, the (201) orientation became pronounced for the sample obtained at the temperature in the reactor deposition zone of 250 °C (Figure 9b) and was slightly reduced with the increase in temperature from 250 to 350 °C (Figure 9c). As already shown in Section 3.1, the presence of this peak is characteristic of the elongated ZnO structures.

The Raman spectra for the ZnO structures obtained at the various temperatures in the reactor deposition zone are shown in Figure 10. Similar to the case described in Section 3.1, three main regions can be distinguished: the typical ZnO peak at about 435 cm^−1^, as well as the peaks at about 275 cm^−1^ and 580 cm^−1^. 

With a decrease in the reactor temperature from 350 to 250 °C, the intensity of all the observed peaks practically did not change; however, at a temperature of 25 °C, the intensity of the peak at about 435 cm^−1^ noticeably increased, and the peaks at about 275 cm^−1^ and 580 cm^−1^ were significantly smoothed out.

Comparing the Raman spectra for two experiments (Figure 6 and Figure 10), it can be concluded that the size of the nanoparticles does not affect their defectiveness and the degree of bond breaking, in contrast with their shape.

### 3.3. Controlling the Size and Morphological Characteristics of ZnO Nanoparticles

The parameter controlling the morphology (shape) of the obtained nanoparticles is the zinc source temperature. Thus, the ZnO powders were formed with spherical, columnar, and rod-like particle shapes at the temperatures of 370, 420, and 470 °C, respectively; this was probably caused by a change in a mechanism of a plasma–chemical reaction. It was also found that the Zn excess in the deposited powder increased with the rise in the zinc source temperature. Based on the Raman spectra, it was shown that the Zn excess most likely led to intensification for the occurrence of structural defects and bond breaking.

The parameter controlling the dimensional characteristics of the obtained nanoparticles is the reactor temperature in the deposition zone. With a decrease in this parameter from 350 to 250 °C, the transverse size of the deposited ZnO particles was reduced by a factor of three. This effect could be associated with an increase in the specific input energy of the process due to the decrease in the temperature causing a fall in the concentration of the plasma-forming particles. Moreover, it is well known that the lattice parameters are temperature dependent; namely, increasing temperature can lead to lattice expansion with a subsequent increase in the resulting structures [23]. However, active cooling of the reactor in the deposition zone was not applicable in this case, since this parameter affected the relaxation rate of the particles at the surface of the powder tank. The lower the temperature, the more equilibrium the process would be, and therefore, the more perfect the structure to be deposited. When producing nanopowders, “perfect” does not mean “best”, since such an equilibrium process led to the formation of dense monocrystalline layers, which was observed at the reactor temperature in the deposition region equal to 25 °C. It was also found that the size of the nanoparticles did not affect their defectiveness and the degree of bond breaking, in contrast with their shape.

It is also worth noting that an additional process parameter that controls the dimensional characteristics of the obtained nanoparticles can be the plasma discharge power as was shown in the previous work of the authors [34]. In that work, it was shown that the transverse diameter of the ZnO rod-like structures decreased by 30 times (from 900 to 30 nm) with an increase in the plasma discharge power from 30 to 70 W. 

## 4. Conclusions

The direct one-stage synthesis of the ZnO nanoparticles was carried out by the PECVD method from elemental high-purity zinc in the zinc–oxygen–hydrogen plasma-forming mixture with the variable deposition parameters.

The dimensional and morphological parameters of the obtained ZnO powders were studied, as well as their structural properties and homogeneity. The study was carried out using methods such as SEM, XRD, and Raman spectroscopy, as well as statistical methods for processing and analyzing experimental data.

In order to determine the optimal parameters for the PECVD synthesis of the ZnO nanoparticles, a series of experiments were carried out. In each of them, one operating parameter was changed while the rest were constant.

It was found that the zinc source temperature was a parameter controlling the morphology (shape) of the obtained nanoparticles, and the reactor temperature in the deposition zone was a parameter controlling their dimensional characteristics. An additional parameter controlling the dimensional characteristics of the nanoparticles was the plasma discharge power.

Based on the analysis of the obtained experimental data, it can be concluded that, in order to obtain ZnO nanoparticles with given size and morphological characteristics in the PECVD process, it is necessary (1) to increase the zinc source temperature to obtain more elongated structures in one direction (and vice versa), (2) to increase the plasma discharge power for reducing the transverse size of the deposited structures (and vice versa), and (3) to lower the reactor temperature in the deposition zone to reduce the transverse size of the deposited structures (and vice versa). However, take into account that at relatively low temperatures instead of powder ones, film structures can form.

## Figures and Tables

**Figure 1 nanomaterials-12-01838-f001:**
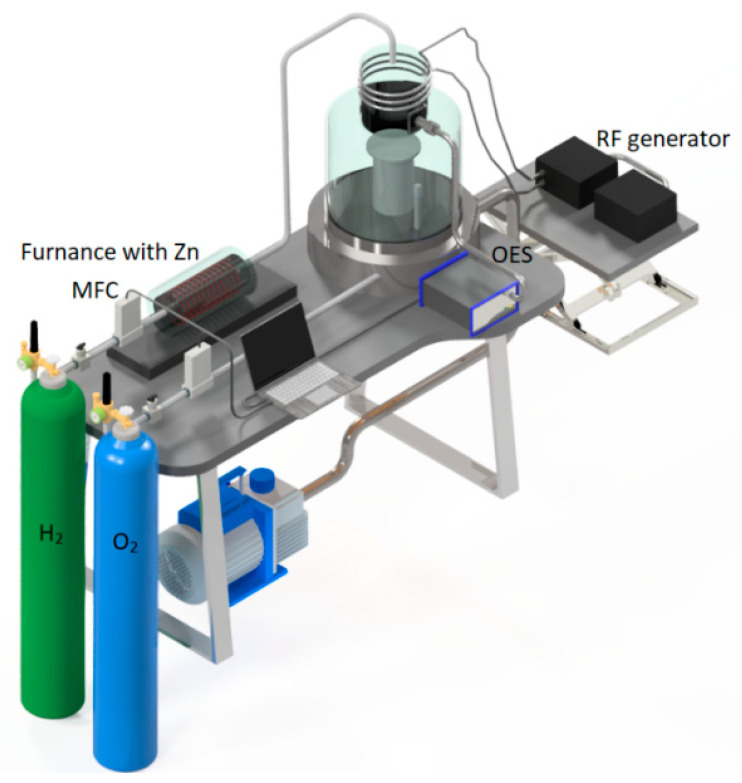
The scheme of a plasma–chemical installation with a pear-shaped reactor.

**Figure 2 nanomaterials-12-01838-f002:**
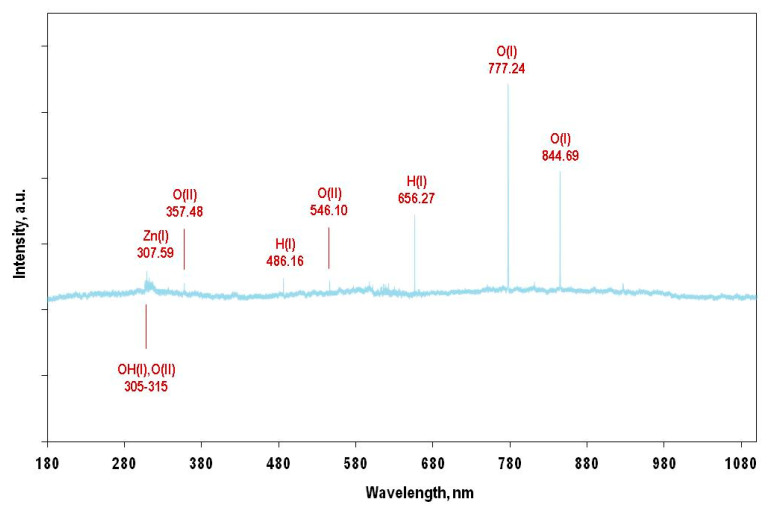
The optical emission spectra of the zinc–hydrogen–oxygen plasma (Zn:H_2_:O_2_ = 2:1:1).

**Figure 3 nanomaterials-12-01838-f003:**
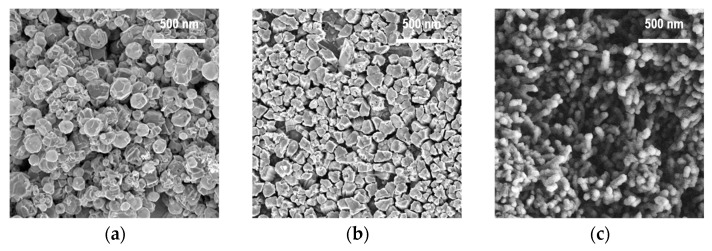
The SEM images of the ZnO nanopowders obtained by the PECVD method at the various zinc source temperatures: (**a**) 370 °C, (**b**) 420 °C, (**c**) 470 °C.

**Figure 4 nanomaterials-12-01838-f004:**
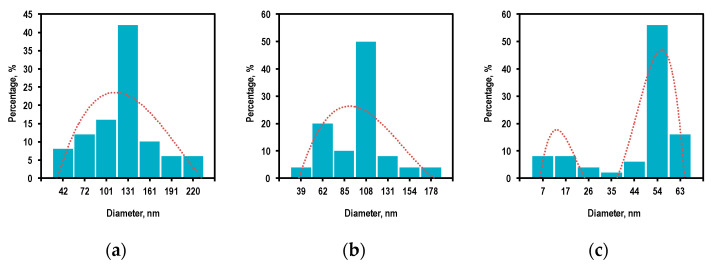
The transverse size distribution of the ZnO structures obtained by the PECVD method at the various zinc source temperatures: (**a**) 370 °C, (**b**) 420 °C, (**c**) 470 °C.

**Figure 5 nanomaterials-12-01838-f005:**
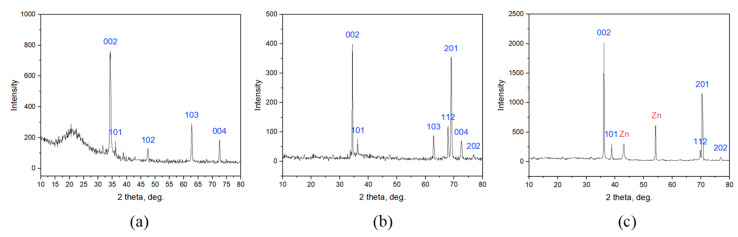
The XRD analysis results for the ZnO structures obtained by the PECVD method at the various zinc source temperatures: (**a**) 370 °C; (**b**) 420 °C; (**c**) 470 °C.

**Figure 6 nanomaterials-12-01838-f006:**
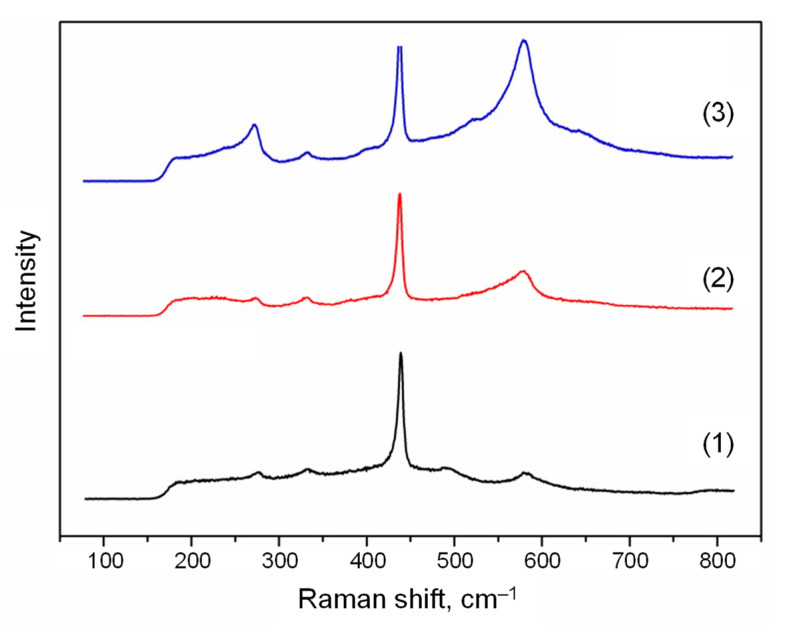
The Raman spectra of the ZnO structures obtained by the PECVD method at the various zinc source temperatures: (1) 370 °C, (2) 420 °C, (3) 470 °C.

**Figure 7 nanomaterials-12-01838-f007:**
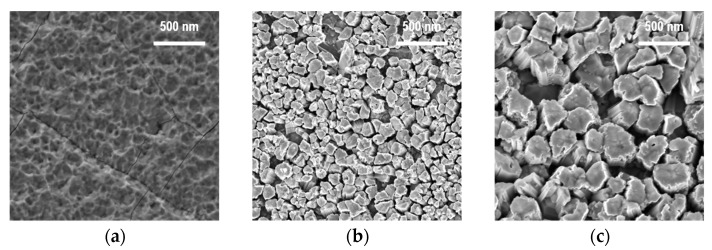
SEM images of ZnO nanopowders obtained by the PECVD method at various temperatures in the reactor deposition zone: (**a**) 25 °C, (**b**) 250 °C, (**c**) 350 °C.

**Figure 8 nanomaterials-12-01838-f008:**
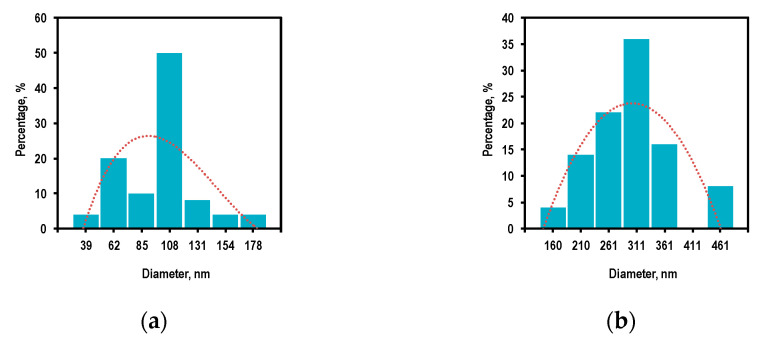
The transverse size distribution of the ZnO structures obtained by the PECVD method at the various temperatures in the reactor deposition zone: (**a**) 250 °C; (**b**) 350 °C.

**Figure 9 nanomaterials-12-01838-f009:**
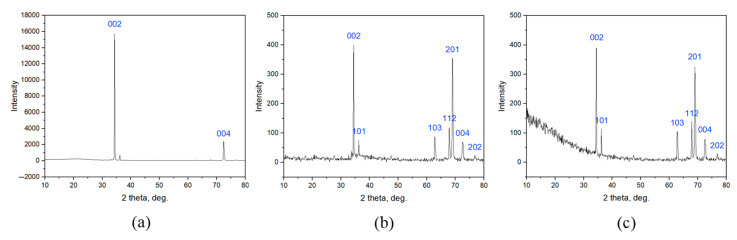
The XRD analysis results for the ZnO structures obtained by the PECVD method at the various temperatures in the reactor deposition zone: (**a**) 25 °C; (**b**) 250 °C; (**c**) 350 °C.

**Figure 10 nanomaterials-12-01838-f010:**
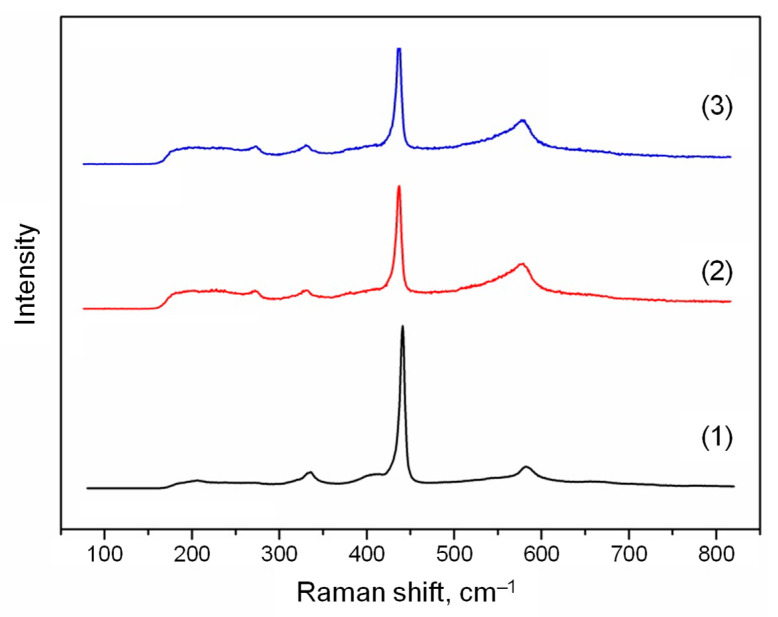
The Raman spectra of the ZnO structures obtained by the PECVD method at the various temperatures in the reactor deposition zone: (1) 25 °C; (2) 250 °C; (3) 350 °C.

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
