# Peer review of "Influence of Temperature Parameters on Morphological Characteristics of Plasma Deposited Zinc Oxide Nanoparticles"

_nanomaterials, 2022, doi:10.3390/nano12111838_

Round 1

Reviewer 1 Report

The manuscript by Sazanova et al presents an interesting study on the operating conditions influencing final structural parameters in the production of ZnO nanostructures by PECVD. As said the study is interesting and methodological, but a few issues follow:

  1. From line 47: having so many examples as mentioned of ZnO synthesis, an example on Pt NPs doesn't make sense here
  2. In line 62: the meaning of one-stage is not clear. Do you mean one-step synthesis? If so, most of the chemical methods are also one-step. An it can be argued that this is not a real one-step as for many of the applications proposed a second step to functionalise/coat the NPs will be required...
  3. Section 3.1: the resolution of the SEM images makes it difficult to see morphology/size. TEM would be more indicated
  4. Fig 3: quantification of sizes from SEM is a bit complicated for the same reason as above. Also in non-spherical samples, what dimension is reported as diameter? Two dimensions should probably be reported (length and diameter).
  5. Fig 4: similar problem with the AFM. Resolution is an issue. Also in this case depth is an issue. For example sample c) does not look like rods in AFM
  6. line 178: 450 ºC is not on the table
  7. Section 3.2: in this case it would be interesting to find the T limit at which the transition between planar to NPs happens
  8. Line 285 mentions fig 2 when it should be fig 11
  9. In several points along the draft the authors assign signals to processes without reference. For example in the description of the Raman, 'According to the Raman spectra (Fig. 6), the excess Zn content, most likely, leads to the formation of point defects and bond breaking during the agglomeration of ZnO nano-particles. This process causes the formation of a shoulder in the region of 100-300 cm-1 and a peak at about 580 cm-1 along with the peak typical for this material at about 435 cm-1.' How do they relate point defect or bond breaking to the peaks? At the minimum references are needed to support this.
  10. Conclusions: the parameters studied were tested independently. But do they really work independently? the authors should test their conclusions by making bespoke structures controlling/modifying all parameters at the same time

Author Response

Dear Reviewer,

We would like to thank you very much for the attention paid to our manuscript and priceless scientific comments. We appreciate a significant work done on this manuscript with detailed analysis that was very valuable in order to change the text and improve the manuscript. The reviewer comments were attentively examined and answers were compiled with the relevant amendments to the text of the paper.

The responds to your comments are as follows:

  1. From line 47: having so many examples as mentioned of ZnO synthesis, an example on Pt NPs doesn't make sense here.

Your advice is great, and no doubt it will be of great value in improving the accuracy of the manuscript.

We made the following revision to the manuscript. “For example, Hammad T. and co-workers [11] reported changing a red-shift from 3.62 to 3.33 eV when the average particle size of ZnO nanoparticles was increased from 11 to 87 nm, respectively. In another work, Mornani E. and co-workers [12] reported a change in the band gap from 4.45 to 4.08 eV with an increase in the average size of ZnO nanoparticles from 46 to 66 nm, respectively.”

  1. In line 62: the meaning of one-stage is not clear. Do you mean one-step synthesis? If so, most of the chemical methods are also one-step. And it can be argued that this is not a real one-step as for many of the applications proposed a second step to functionalise/coat the NPs will be required...

Thank you for your comment.

Plasma initiation makes it possible to eliminate the pollution possibility of the final product with equipment materials and to achieve 100% conversion of initial substances due to establishing kinetic dependencies during the course of plasma-chemical reactions. In this context, PECVD does not require additional purification steps for the final product, so it can be classified as a one-step synthesis method.

Appropriate clarifications were included in the text of the manuscript.

  1. Section 3.1: the resolution of the SEM images makes it difficult to see morphology/size. TEM would be more indicated.

We gratefully appreciate for your valuable suggestion.

Unfortunately, we cannot provide the TEM data for we don’t have the appropriate device. However, as a compromise, we replaced the SEM images provided in the manuscript with higher magnification ones.

  1. Fig 3: quantification of sizes from SEM is a bit complicated for the same reason as above. Also in non-spherical samples, what dimension is reported as diameter? Two dimensions should probably be reported (length and diameter).

Thanks for your constructive comment.

The size of non-spherical samples was measured as the diameter of their cross-section. For additional control of the measurement results, also the analysis of the SEM images (determination of the average size of equivalent disk (Davg) of NPs cross-section) was carried out by the method of watershed segmentation in the SPMLab™ v5 software package.

Appropriate explanations were added to the text of the manuscript.

  1. Fig 4: similar problem with the AFM. Resolution is an issue. Also in this case depth is an issue. For example sample c) does not look like rods in AFM.

We fully agree with your constructive comment.

Unfortunately, higher resolution AFM of nanoparticles requires a special vacuum chamber and the corresponding modes of microscope operation, such as a peak force one. We don’t have the appropriate device. However, as a compromise, we excluded the AFM data from the text of the manuscript.

  1. Line 178: 450 ºC is not on the table.

We’re so sorry that we brought you trouble for our mistake.

The manuscript should have indicated 470 °C instead of 450 °C. The typo was corrected.

  1. Section 3.2: in this case it would be interesting to find the T limit at which the transition between planar to NPs happens.

We completely agree with you, it would really be interesting to establish the T limit at which the transition between planar structure to NPs one happens. However, such a study requires numerous additional experiments and re-equipment of the experimental setup. We hope to conduct a similar study in the future.

  1. Line 285 mentions fig 2 when it should be fig 11.

You are absolutely right. Thank you for your comment.

We were taken this into account. However, after the revision of the manuscript, the numbering of the figures was changed. Figure 11 became Figure 2.

  1. In several points along the draft the authors assign signals to processes without reference. For example in the description of the Raman, 'According to the Raman spectra (Fig. 6), the excess Zn content, most likely, leads to the formation of point defects and bond breaking during the agglomeration of ZnO nano-particles. This process causes the formation of a shoulder in the region of 100-300 cm-1 and a peak at about 580 cm-1 along with the peak typical for this material at about 435 cm-1.' How do they relate point defect or bond breaking to the peaks? At the minimum references are needed to support this.

That’s really nice comment.

We made the following revision to the manuscript. “The Raman spectra for the ZnO structures obtained at the various zinc source temperatures are shown in Figure 5. The peak at about 435 cm-1 is typical for ZnO structures, while the peaks at about 275 cm-1 and 580 cm-1 do not correspond to ZnO normal modes. These peaks are additional vibrational modes and can be associated with defects and bond breaking, respectively [31–33].”

  1. Conclusions: the parameters studied were tested independently. But do they really work independently? The authors should test their conclusions by making bespoke structures controlling/modifying all parameters at the same time.

We agree that it would be interesting to conduct a cross-sectional experiment with a simultaneous change in all parameters. However, our pre-experimental selection of the parameters showed that such manipulations do not lead to conclusions strikingly different from those drawn in this work. The general trend for the influence of the zinc source temperature on the shape of the NPs, as well as the influence of the temperature in the deposition zone and the plasma discharge power on their size, remains.

We appreciate for your warm work earnestly, and hope that the correction will meet with approval. Once again, thank you very much for your comments and suggestions.

Yours Sincerely,

Dr. Tatyana S. Sazanova.

Reviewer 2 Report

In this manuscript, the authors reported the synthesis of zinc oxide nanoparticles using plasma-enhanced chemical vapor deposition (PECVD) method. They used XRD, SEM, AFM and Raman spectra to characterize samples, and studied the influence of zinc source temperature, and a reactor temperature in a deposition zone on the size and morphology. I do not recommend publish this manuscript in this journal for the following reasons:

  1. In the first paragraph ofintroduction, the authors ascribed large specific surface area and pore volume to the advantage of Zinc oxide. In my opinion, large specific surface area and pore volume were determined by the structure and morphology, not the material was zinc oxide.
  2. The authors said PECVD provides cost-effectiveness, high purity of the resulting material, while, this method was not cost-effectiveness as shown in Figure 1 and the samples obtained in this report were not very pure.
  3. All the characterization in this manuscript were discussed very rough and they should discuss the results in detail.
  4. HR-SEM images should be provided to present the morphology more clearly; the PDF cards of different phases that maybe present in the measured samples should be added in the XRD patterns; the peaks in the Raman spectra should be ascribed.
  5. The average size of the samples decreased with the increasing temperature, the authors should give explanation.
  6. How can the authors judge the samples were single-crystal or polycrystalline from XRD patterns?
  7. The possible mechanisms of plasma-chemical reactions should be consolidated by more methods.
  8. The English should be polished.

Author Response

Dear Reviewer,

On behalf of my co-authors, we are very grateful to you for giving us an opportunity to revise our manuscript. We appreciate you very much for your constructive comments and suggestions on our manuscript. We have studied reviewer’s comments carefully and tried our best to revise our manuscript according to the comments. The following are the responses and revisions we have made in response to the reviewer’s questions and suggestions on an item-by-item basis. Thanks again to the hard work of you!

The responds to your comments are as follows:

  1. In the first paragraph of introduction, the authors ascribed large specific surface area and pore volume to the advantage of Zinc oxide. In my opinion, large specific surface area and pore volume were determined by the structure and morphology, not the material was zinc oxide.

Thanks for your constructive comment.

We fully agree with it. Indeed, a large specific surface area and pore volume are determined by the structure and morphology of obtained materials.

Appropriate changes were made to the text of the manuscript.

  1. The authors said PECVD provides cost-effectiveness, high purity of the resulting material, while, this method was not cost-effectiveness as shown in Figure 1 and the samples obtained in this report were not very pure.

Thank you for your comment.

Plasma initiation makes it possible to significantly reduce the temperature of reactor walls and a deposition zone as well as to eliminate the pollution possibility of the final product with equipment materials and to control the deposition zone temperature over a wider range, thus setting the conditions for structure growth. Plasma initiation also makes it possible to achieve 100% conversion of initial substances due to establishing kinetic dependencies during the course of plasma-chemical reactions. In this context, PECVD is cost efficient and provides high purity of resulting materials.

Appropriate clarifications were included in the text of the manuscript.

  1. All the characterization in this manuscript were discussed very rough and they should discuss the results in detail.

Thank you for this valuable notice.

We revised the description of the results and made the appropriate changes to expand their discussion.

  1. HR-SEM images should be provided to present the morphology more clearly; the PDF cards of different phases that maybe present in the measured samples should be added in the XRD patterns; the peaks in the Raman spectra should be ascribed.

We gratefully appreciate for your valuable suggestion.

Unfortunately, we cannot provide the HR-SEM data for we don’t have the appropriate device. However, as a compromise, we replaced the SEM images provided in the manuscript with higher magnification ones.

In Section 2.5 “X-ray structural analysis”, the information of the PDF cards for ZnO and Zn were added.

The analysis of the peaks in details was added to the Raman spectra description.

  1. The average size of the samples decreased with the increasing temperature, the authors should give explanation.

The average size of the samples decreased with the increasing temperature of the zinc source, as it is assumed, since the plasma-chemical mechanism of the process changes.

Probably, zinc becomes more active and more readily "allows" an electron to be detached from it with an increase in its initial temperature, transverses to ionic form, behaves more selectively in combining with oxygen ions, and, as a result, is less prone to agglomeration.

Moreover, such change in the transverse size of the nanoparticles can be associated with the corresponding redistribution of Zn and O atoms in spherical and elongated crystal structures.

Appropriate clarifications were included in the text of the manuscript.

  1. How can the authors judge the samples were single-crystal or polycrystalline from XRD patterns?

Thanks for your constructive comment.

We fully agree with it. To determine the type of NPs’ crystallinity, the use of XRD is not sufficiently appropriate, since different orientations of particles can introduce corresponding changes in XRD signals.

For nanoparticles, there can be three different cases: (1) the orientation of single-crystal nanoparticles is different; (2) the orientation of polycrystalline nanoparticles is different; (3) all polycrystalline nanoparticles are equally oriented. Thus, from the XRD data, we can only state that the nanoparticles in our case have a wurtzite-type crystal structure.

Appropriate changes were made to the text of the manuscript.

  1. The possible mechanisms of plasma-chemical reactions should be consolidated by more methods.

That’s really nice comment.

We fully agree with it. For this reason, we used quantum chemical calculations by the density functional theory (DFT) in tandem with the method of optical emission diagnostics.

Optical emission spectroscopy makes it possible to detect the presence of chemically active excited particles in plasma from characteristic emission lines. Next, a set of more than 70 possible reactions is built. As a result of quantum-chemical modeling, the most probable reactions are calculated taking into account the lifetime of excited particles and the rate constants of plasma-chemical reactions.

Nerveless, on the advice from one of the reviewers, we excluded the part concerning the discussion of the possible mechanisms for the plasma-chemical process from the text of the manuscript.

  1. The English should be polished.

We gratefully appreciate for your valuable comment.

We revised the manuscript in order to polish and improve English.

We sincerely hope that this revised manuscript has addressed all your comments and suggestions. We appreciated for your warm work earnestly, and hope that the correction will meet with approval. Once again, thank you very much for your comments and suggestions.

Yours Sincerely,

Dr. Tatyana S. Sazanova.

Reviewer 3 Report

This appears to be a well thought out and executed series of synthesis experiments.  Therefore the results if reported properly and adequately could be of general interest.  However, there are big problems in the reporting that will require major revision. 

The biggest problem is with the reported EDS results.  They give the totally incorrect impression that there are large variations in the stoichiometry of the ZnO produced in the different runs.  That is not possible if the product is crystalline ZnO as the x-ray data demonstrate it is, even in the one case where some Zn metal was also deposited.  It is well known that the apparent EDS compositions for particulate samples depend strongly on particle size, shape, and orientation, and that it is extremely difficult to compensate for these influences.  Even under the best circumstances it is not possible for EDS analyses to be accurate enough to detect any possible real difference in stoichiometry between samples.  I think it is best not to report these EDS results at all.  I do not think they can be made accurate enough to be useful.

It is explicitly stated that the samples for AFM analyses were prepared by depositing thin layers of the prepared powders on conductive tape.  Although it is not stated, and it should be, how the samples SEM, EDS, XRD, and Raman were prepared it seems reasonable to assume they were prepared in the same way.

It is not useful to state that objective lens used in the Raman spectrometer had “an aperture of 0.45”.  What would be informative is to say what size the laser beam spot was on the sample.  I am guessing that it would have had a diameter of about 5 micrometers so that a spectra are an average over many grains.  This is relevant since the Raman spectra are expected to depend on crystal orientation.

Since the big variations in stoichiometry inferred from EDS are not a reality, the interpretation of changes in Raman spectra in terms of them is baseless.  The differences in Raman spectra more likely just reflect differences in average grain orientation in the samples.

Just presenting the experimental results including the AFM, SEM, and XRD information on the size and shape of the particles should make a good paper.  The Raman results are also fine as long as they are not over interpreted.  The models for ZnO formation in the plasma also seem irrelevant to the actual nucleation and growth of ZnO crystals.

Author Response

Dear Reviewer,

Thank you very much for your kindly comments on our manuscript. There is no doubt that these comments are valuable and very helpful for revising and improving our manuscript. In what follows, we would like to answer the questions you mentioned and elaborate on the changes made to the original manuscript.

The responds to the reviewer’s comments are as follows:

  1. The biggest problem is with the reported EDS results. They give the totally incorrect impression that there are large variations in the stoichiometry of the ZnO produced in the different runs. That is not possible if the product is crystalline ZnO as the x-ray data demonstrate it is, even in the one case where some Zn metal was also deposited. It is well known that the apparent EDS compositions for particulate samples depend strongly on particle size, shape, and orientation, and that it is extremely difficult to compensate for these influences.  Even under the best circumstances it is not possible for EDS analyses to be accurate enough to detect any possible real difference in stoichiometry between samples.  I think it is best not to report these EDS results at all.  I do not think they can be made accurate enough to be useful.

Thank you for this valuable comment.

We fully agree with it. Indeed, it is difficult to judge the real differences in stoichiometry from the data of the EDS analyses. In this regard, we decided to exclude the appropriate data to avoid misunderstanding.

  1. It is explicitly stated that the samples for AFM analyses were prepared by depositing thin layers of the prepared powders on conductive tape. Although it is not stated, and it should be, how the samples SEM, EDS, XRD, and Raman were prepared it seems reasonable to assume they were prepared in the same way.

We gratefully appreciate for your valuable notice.

Information about sample preparation was added to the appropriate parts of the manuscript.

  1. It is not useful to state that objective lens used in the Raman spectrometer had “an aperture of 0.45”. What would be informative is to say what size the laser beam spot was on the sample.  I am guessing that it would have had a diameter of about 5 micrometers so that a spectra are an average over many grains.  This is relevant since the Raman spectra are expected to depend on crystal orientation.

Thanks for your constructive comment.

You're right that the shape of Raman spectra depends on crystal orientation. However, in our case, based on the XRD data, the preferred orientation of the nanoparticles for all samples was (002). Moreover, for the sample obtained at the source temperature of 420 °C (Fig. 5b), the (201) orientation became pronounced, and the peak intensities in the Raman spectra in the region of 100-300 cm-1 and 580 cm-1 had average values relative to other samples. The maximum intensities of these peaks were reached in another sample obtained at the source temperature of 470 °C, when the height of the (201) reflection on XRD was already reduced. Thus, we believe that changes in the form of the Raman spectra are associated precisely with defects such as interstitial zinc and bond breaking. Appropriate explanations were added to the text of the manuscript.

However, we fully agree that the information about the spot size of the laser beam is important, so it was also added to the manuscript.

  1. Since the big variations in stoichiometry inferred from EDS are not a reality, the interpretation of changes in Raman spectra in terms of them is baseless. The differences in Raman spectra more likely just reflect differences in average grain orientation in the samples.

That’s really nice comment.

The analysis of the peaks in terms of crystal orientation was added to the Raman spectra description.

  1. Just presenting the experimental results including the AFM, SEM, and XRD information on the size and shape of the particles should make a good paper. The Raman results are also fine as long as they are not over interpreted. The models for ZnO formation in the plasma also seem irrelevant to the actual nucleation and growth of ZnO crystals.

We value your professional opinion.

We decided to take your advice and excluded the part concerning the discussion of the possible mechanisms for the plasma-chemical process from the text of the manuscript.

Thank you again for your positive and constructive comments and suggestions on our manuscript. We hope you will find our revised manuscript acceptable for publication.

Yours Sincerely,

Dr. Tatyana S. Sazanova.

Round 2

Reviewer 1 Report

It is clear that the authors have made an effort to improve the overall quality of the manuscript. Even though some of the questions raised remain unanswered, I believe that the manuscript in its current form can be published.

Reviewer 2 Report

The authors has answered all my concerns. And I recommend to publish this work. Best

Reviewer 3 Report

The authors have done a good job of rapidly making the changes I requested.